# Inequalities in Health Care Experience of Patients with Chronic Conditions: Results from a Population-Based Study

**DOI:** 10.3390/healthcare9081005

**Published:** 2021-08-05

**Authors:** Roberto Nuño-Solínis, Maider Urtaran-Laresgoiti, Esther Lázaro, Sara Ponce, Juan F. Orueta, María Errea Rodríguez

**Affiliations:** 1Deusto Business School Health, University of Deusto, 48014 Bilbao, Spain; roberto.nuno@deusto.es; 2Research Group in Social Determinants of Health and Demographic Change (OPIK), University of the Basque Country (EHU-UPV), 48940 Leioa, Spain; maider.urtaran@ehu.es; 3Faculty of Health Sciences, Valencian International University, 46002 Valencia, Spain; melazaro@universidadviu.com; 4International Research Projects Office, University of Deusto, 48007 Bilbao, Spain; sponce@hotmail.com; 5Primary Health Care Center of Astrabudua, OSI Uribe, Osakidetza Basque Health Service, 48950 Erandio, Spain; jon.orueta@osakidetza.net; 6Independent Researcher, 31007 Pamplona, Spain

**Keywords:** chronic conditions, inequalities, patients’ health care experience, multimorbidity, quality of life, population-based study

## Abstract

Patients’ experience is an acknowledged key factor for the improvement of healthcare delivery quality. This study aims to explore the differences in healthcare experience among patients with chronic conditions according to individual sociodemographic and health-related variables. A population-based and cross-sectional study was conducted. The sample consisted of 3981 respondents of the Basque Health Survey (out of 8036 total respondents to the individual questionnaire), living in the Basque Country, aged 15 or older, self-reporting at least one chronic condition. Patient experience was assessed with the Instrument for Evaluation of the Experience of Chronic Patients questionnaire, which encompasses three major factors: interactions between patients and professionals oriented to improve outcomes (productive interactions); new ways of patient interaction with the health care system (the new relational model); and the ability of individuals to manage their care and improve their wellbeing based on professional-mediated interventions (self-management). We conducted descriptive and regression analyses. We estimated linear regression models with robust variances that allow testing for differences in experience according to sociodemographic characteristics, the number of comorbidities and the condition (for all chronic or for chronic patients’ subgroups). Although no unique inequality patterns by these characteristics can be inferred, females reported worse global results than males and older age was related to poorer experience with the new relational model in health care. Individuals with lower education levels tend to report lower experiences. There is not a clear pattern observed for the type of occupation. Multimorbidity and several specific chronic conditions were associated (positive or negatively) with patients’ experience. Health care experience was better in patients with greater quality of life. Understanding the relations among the patients’ experience and their sociodemographic and health-related characteristics is an essential issue for health care systems to improve quality of assistance.

## 1. Introduction

Chronicity poses a serious challenge for health systems worldwide as it is rising as a consequence of the aging of populations and other well-known factors associated with unhealthy lifestyles [1]. Chronic conditions rarely occur alone [2]. Consequently, and adding to this the aging of the population, multimorbidity is rising [3,4]. Having multiple comorbidities has been associated with poorer health outcomes [5,6]. 

New health care models have been developed to respond to the needs of patients with chronic conditions [7] and improve their experiences of care [8], by addressing fragmentation and putting emphasis on patients’ narratives, preferences, and expectations [9,10]. 

The Basque Health Service provides public health care to over two million inhabitants in the region, around 22% of them aged 65 or over [11]. The Basque Government developed a Strategy for Tackling the Challenge of Chronicity in 2010 [12]. It contained policies and projects aimed at reinventing the health delivery model to improve the quality of care for patients with chronic conditions and advance towards a more sustainable, proactive, and integrated model. 

There is an extensive list of papers published on the influence of sociodemographic factors, chronic conditions, and multimorbidity on patient-reported experience with health care services. Patient experience entails looking for information provided by patients on their interactions with health and social professionals and results obtained from those interactions [8]. This is different from patient satisfaction, which refers to patients’ emotions, feelings, and their perception of delivered health services [13,14]. Patient experience is increasingly acknowledged as a key aspect of quality health care delivery; its measurement is being developed through patient-reported experience measures (PREMs) and it is increasingly recognized in performance frameworks, such as the Quadruple Aim [15]. Moreover, there is evidence that patient experience is linked to safety and clinical effectiveness outcomes [16]. Studies report opposing findings on patient experience with some chronic conditions according to sociodemographic characteristics, including social and educational level [17,18]. Evidence has also found that experiences of patients with multiple long-term conditions are not different from patients with a single long-term disease [19]. However, multimorbidity is generally associated in the literature with poorer health outcomes [5,6], including poorer health care experience [20]. Providers tend to face, as a result, extra challenges when managing these patients [21]. 

As the measurement of patient experience is considered to be a key element in determining the quality of health care and social care provided, it is relevant to analyze the factors and patients’ characteristics that influence this experience. Previous studies have found that better experiences with health care services are more likely to be reported by patients with higher education levels or who are more affluent [22], and among women [23]. Other studies have found opposite results for gender [24,25,26] or education [17]. Furthermore, there is evidence of the negative impact on the experience of care of patients with long-standing conditions [27], particularly observed for mental health [25,27] and cancer [25]. Variation in patient experience by cancer type has been observed in a nationwide study [28]. There seems to be an association between quality of life and patients’ experience with health care factors. However, the association between individual sociodemographic and health-related characteristics on patients´ experience for those suffering from chronic conditions remains inconclusive. For example, a study found patients with HIV, with more complexity of clinical care, reported lower quality of life, which was explained by negative experiences during productive interactions (interactions between patients and professionals oriented to improve outcomes) and self-management factors (the ability of individuals to manage their care besides services provided by health professionals) [29]. Evidence has also shown how in patients with long-standing diseases with good disease control, functional limitations impact quality of life (QoL) and health-related quality of life (HRQoL), which appear to be inversely correlated with the self-management factor [30]. More evidence from population-based studies to clarify these associations is needed.

Few questionnaires validated to assess patients’ experiences cover all dimensions and concepts of health care comprehensively [31,32]. The IEXPAC instrument (available at http://www.iexpac.org, accessed on 5 January 2021) [8] can detect differences between patient subgroups. In addition to information about patients’ clinical and risk factors, it is an important piece of information for decision-makers [33]. IEXPAC introduces a new focus on the interaction between patients and health care teams through the use of new technologies and patient-to-patient interactions [26]. It incorporates a broader notion of integrated care, including social care and patient self-management [26].

Understanding how patients with chronic conditions experience the health care process is relevant for the design of interventions better tailored to address the needs of the increasing number of individuals with these conditions [34]. We, thus, designed this study.

We examine the association between a set of experience factors and sociodemographic and health-related characteristics in a sample representative of the population of one Spanish region (Basque Country). The Basque Health survey included the Instrument for Evaluation of the Experience of Chronic Patients (IEXPAC) questionnaire as a means for evaluating the experience of care of these patients. IEXPAC has been used in several studies for analyzing the influence of demographic and health care-related variables on patient experiences [26,35,36]. However, few studies have been undertaken in a representative population, most of them being conducted in selected subpopulations [36,37].

We want to test if the level of experience with health care services of patients with self-declared chronic conditions is associated with: The individual sociodemographic characteristics.The number of chronic conditions declared by the individual.The chronic condition(s) declared.The individual level of self-reported quality of life.

We conduct descriptive and regression analyses to test these hypotheses and discuss the implications of our analyses.

## 2. Materials and Methods

### 2.1. The Basque Health Survey

The Basque Health Survey (ESCAV) is conducted by the Basque government every five years with people living in the Basque Country. The 2018 survey employed a representative sample of 5300 households that included an individual questionnaire with more than eight thousand final responses (*n* = 8036). This sample of people surveyed is representative of the Basque Country population. 

The aims, methodology, and sampling are explained elsewhere (https://en.eustat.eus/document/encsalud_i.html, accessed on 5 January 2021). 

### 2.2. Ethical Considerations

The study was conducted per the authorization of the Basque Government, as the owner and institution responsible of the data, for the use of the database, and it was approved by the University of Deusto Ethics Committee (ETK-24/20-21).

### 2.3. Design and Working Sample

We conducted a cross-sectional study which analyzes data obtained from the Basque Health Survey 2018. The survey includes self-reported sociodemographic and economic data, diseases, self-assessed quality of life, and experience of care. Because our study population consists of all the Basque Country inhabitants aged 15 or older, who declared in the survey having at least one chronic condition that had been diagnosed by a medical doctor, our sample includes nearly 50% of the total respondents of the individual questionnaire (*n* = 3891 individuals).

### 2.4. Main Study Variables

#### 2.4.1. Dependent Variables

To assess the experience with the health care delivery process we used the IEXPAC instrument. The IEXPAC questionnaire contains 11 individual items, to which the patients answer on a five-point Likert scale. As dependent variables, we employed three IEXPAC factors and the global IEXPAC score, derived from the combination of the responses to the 11 items. The three factors (generated and provided with the dataset) and overall IEXPAC score take values between 0 and 10, a score of 0 representing the worst possible reported experience and a score of 10 representing the best possible reported experience. The three factors were: 

Factor (1) Productive Interactions (INTER): composed of responses to IEXPAC items 1, 2, 5, and 9, and represents the characteristics and content of interactions between patients and professionals oriented to improve outcomes. 

Factor (2) The New Relational Model (NEW): items 3, 7, and 11. Represents new ways of patient interaction with the health care system, through the internet or with peers.

Factor (3) Patient Self-Management abilities (SELF): items 4, 6, 8, and 10. Represents the ability of individuals to manage their care and improve their wellbeing based on professional-mediated interventions. 

Finally, the overall experience was measured by the IEXPAC score (OVERALL IEXPAC). For a full description and meaning of each of the items in the IEXPAC, please refer to Figure 1.

#### 2.4.2. Independent Variables

Sociodemographic and economic characteristics: gender, age, level of education, income, and occupation.
Gender.Age. We based our classification on the one used by the National Institute for Statistics in Spain (INE). The INE uses quinquennial age ranges (0–4, 5–9, 10–14, 15–19, 20–24, …). We rearranged some of the age groups to have a large enough sample size in each age category (to ensure there is a minimum of 2% of respondents in each range and avoid multicollinearity problems). We, for example, aggregated 15–19 and 20–24 into 15–24. Additionally, for the sample of respondents aged over 65 (our population of retired respondents), we created three groups: we merged 65–69 with 70–74, and 75–79 with 80–84 and 85–89, ensuring this way a similar number of responses in 65–74 and 75–89. We leave the aged 90 or over category separate, as in the INE classification. We aimed to compare these three groups, to test the effect of diseases and multimorbidity and see if, among the retired, there is an aging effect of conditions and/or multimorbidity.The level of education was categorized in the health survey as no education, primary education, lower secondary education, higher secondary education, and tertiary education.Income was categorized in income ranges of net household income.Occupation was categorized according to the National Classification of Occupations, which came into force in 2011 based on the Social Determinants Working Group of the Spanish Society of Epidemiology’s [27] proposal. This classification groups occupational social classes into five groups: managers of companies of highly educated employees (Managers I), managers of companies of less-educated employees (Managers II), intermediate occupations or freelancers (Intermediate), supervisors or technical positions at qualified or semi-qualified occupations (semi-qualified), and supervisors or technical position at non-qualified occupations (non-qualified). Note that occupation might be current (for people of working age) or past (for retired individuals), and the same categories apply for both types of respondents.Health-related variables include chronic conditions and the number of chronic conditions.
Chronic conditions. All 39 chronic conditions presented in the Basque Health Survey were included as dummy variables. An individual can declare that they suffer from any number of conditions on the list at the time of the survey.Number of chronic conditions. This variable was created based on the number of reported chronic conditions. We created ranges: one chronic condition, two chronic conditions, three chronic conditions, and more than three chronic conditions.The EQ-5D utility index is the QoL measure used, obtained based on responses to the EQ-5D-5L instrument based on Ramos-Goñi et al. [38]. This index is validated and accepted as a population-based QoL norm [39].

### 2.5. Data Analysis

We started by conducting a descriptive analysis followed by regression analysis. We looked at one-way and two-way tables, correlations and significant differences between variables’ categories on the mean observed experience with the three health factors and with the global experience seeking care. Descriptive statistics helped us select the explanatory variables. Then, we estimated regression models to test our study hypotheses. Statistical analyses were conducted using Stata SE software.

The regression models are: IEXPAC_fi_ = α_0_ + α_k_ × X_ki_ + ε_i_(1)
IEXPAC_fi_ = β_0_ + β_k_ × X_ki_ + β_j_ × numberCD_ji_ + ε_i_(2)
IEXPAC_fi_ = α_0_ + α_k_ × X_ki_ + α_j_ × CD_ji_ + ε_i_(3)
IEXPAC_fi_ = β_0_ + β_k_ × X_ki_ + β_j_ × EQ-5Dindex + ε_i_(4)
where f is a vector of dependent variables, the IEXPAC factors (INTER, NEW, SELF) and the IEXPAC global (OVERALL IEXPAC); X_ki_ is the vector of *k* sociodemographic variables that describe the i-individual, included as dummies; numberCD_ji_ is the number of j chronic conditions for the i-individual; CD_ji_ is the vector of j = 39 chronic conditions; EQ-5Dindex_i_ is the utility associated with the declared EQ-5D-5L health status for each respondent; and ε_i_ is the error term of the models.

Additionally, regarding our models: (1)The sociodemographic characteristics model, which tests differences in experience according to sociodemographic characteristics of the sample.(2)The multimorbidity model, which shows the average effect of having a certain number of comorbidities (2, 3, or 3+), controlling for sociodemographic factors.(3)The chronic conditions model, which shows the specific effect of each chronic condition on experience with the different factors. This model also offers the opportunity to test the effect of chronic conditions for specific subgroups of patients.(4)The Quality of Life model (QoL), which aims to understand the association between QoL and the experience of this population.

Although the four models provide separate pieces of information, they should be seen as complementary. Similarly, descriptive and regression analyses are also complementary. 

Two- and three-way interactions between age, socioeconomic status (proxy by occupation), and education were included to test whether the effect of socioeconomic status varied between age groups. Interactions between the number of chronic conditions and (1) age ranges and (2) the number of chronic conditions or severity declared were also included to test if there were differences in the experiences of patients between age groups or those with multiple chronic conditions. Interactions between gender and other sociodemographic characteristics were tested but produced insignificant effects of minimal added value, so these were omitted from the model. Note that the QoL model does not include the chronic conditions nor the number of conditions to avoid multicollinearity in our estimations. We used a confidence level of at least 95% in our analyses. 

We tested and corrected the model for heteroscedasticity using heteroscedasticity-consistent standard errors (Eicker–Huber–White standard errors). This implies weighting the variances–co-variances matrix. This method, known as weighted least squares (WLS), makes the variance of the model robust and significantly reduces the bias of heteroskedastic ordinary least squares (OLS) estimators. We did not use any imputation method to replace missing data given the low proportion of missing responses. Statistical analyses were conducted using Stata SE software.

## 3. Results

### 3.1. Descriptive Results

Responses’ distribution to the IEXPAC module are represented in Figure 1 above. The distribution of the study population according to sociodemographic variables (gender, age, net monthly income, level of education, and class/occupation) is shown in Table 1 below. Among the 3891 patients, 55.7% were women. The mean age of the sample was 62.91 years.

In Figure 1 we observe that more than 70% of the patients responded “always” or “mostly” to the items that related to productive interactions (Factor 1, INTER). For all items relating to the new relational model (Factor 2, NEW), around 10% responded “always” or “mostly”. More than 60% for most items of the self-management abilities factor (Factor 3, SELF), except for item 10, where the percentage was below 40%. Consequently, Factor 1 received the highest score in mean in the sample (7.556) and Factor 2 the lowest one, with a mean value of 1.277.

Globally, the sample gives a mean value to the health care-seeking experience of 5.468 points according to the global IEXPAC score. The means and standard deviations for the IEXPAC factors and global IEXPAC score are also shown for each sociodemographic variable, by category. Looking at mean QoL by sociodemographic characteristics we observe slightly lower self-reported health for women, although this difference is not statistically significant. The EQ-5Dindex score decreases also with age but increases with income and education or is higher for most qualified occupations. Some significant differences in experience are observed for the different factors.

There were seven missing responses for occupation, five missing answers for the number of conditions, and one individual who did not respond to the IEXPAC, generating one missing answer for the IEXPAC factors. 

The following table (Table 2) shows the distribution of chronic conditions (sorted by percentage of individuals reporting each of the health problems) as well as the distribution of patients by the number of chronic conditions (1, 2, 3, or more than 3 diseases). 

We observe how the most prevalent diseases in the sample are hypertension (*n* = 1606, 41.3%), high cholesterol (*n* = 1179, 30.3%), osteoarthritis (*n* = 706, 18.1%) and back pain (*n* = 590, 15.2%). We also observe how the mean QoL, represented by the EQ-5Dindex scores, is lower for patients declaring diabetic foot (EQ-5Dindex = 0.494) or incontinence problems (EQ-5Dindex = 0.532) compared to QoL of patients with more prevalent chronic conditions, represented on the top of the table, and showing high QoL indices. The difference in experience between patients with the most and the least prevalent conditions is statistically significant (*p*-value < 0.01). Some significant differences in experience valuation are also observed for the number of chronic conditions.

### 3.2. Regression Results

The results of the regression models are displayed in tables. Table 3 presents the differences in health care experience according to sociodemographic and economic characteristics; Table 4 according to the total number of chronic conditions in each patient; Table 5 according to the presence of specific chronic conditions; and, finally, Table 6 according to quality of life.

According to Model 1 results, better experience with productive interactions and self-management health care factors is associated with being a man. This is also true for experience with the overall reported experience. Aging reduces experience with the new relational model for the older respondents aged 75 or over, reporting lower values compared to the youngest respondents (baseline, 15–24), and this is significant at a 95% confidence level for the population aged 90 or over. To see it with an example, for individuals aged 90 or over, experience with factors related to the new relational model was 2.464 points lower in the mean than for respondents aged 15–24. In addition, those with lower levels of education report, in mean, lower levels of health care seeking experience with all factors. Regarding occupation, significant effects are observed for the new relational model and overall IEXPAC score experiences. In particular, results show a significant difference between those reporting the most qualified employment (Managers I) compared to the second most qualified (Managers II), with experience scoring higher for the most qualified ones in this case. However, for the self-management factor, a significant difference is observed between the intermediate and most qualified occupations (*p*-value < 0.05), and in this case, respondents from intermediate occupations score higher experience. 

The model also allows testing of different associations between respondents’ combinations of sociodemographic characteristics and experience. For example, one could be interested in testing what happens if we interacted age with occupation (See Appendix A in the Appendix A for detailed results, including coefficients for the interactions). For individuals aged 90 or over who had worked in intermediate occupations, the experience score would be equal to 8.369 − 1.121 − 0.660 + 2.659 = 9.247 (note that 2.659 is the interaction effect—provided as Appendix A—for having had a job for most of one’s working life at intermediate qualified occupations and aged 90 or over at the time of the survey). There is a difference in experience compared to the baseline group—younger individuals at intermediate occupations (1.121 points higher). Additionally, being 90 or older and having worked mostly in non-qualified occupations is associated with a positive experience with the new relational model compared to younger individuals reporting the same type of occupation. Interacting age and education, we observe some significant effects, especially for the secondary-lower and upper levels of education. Compared to the youngest group of respondents, all other age groups report better experience with all factors, except for experience with the new relational model (no significant differences were found for this factor by education). Finally, for the overall experience, the negative effect found for all age groups (if we only looked at the effect of age on its own) is compensated for by the effect of working in Manager II-type occupations, inverting the negative effect to a positive effect, as the coefficients for the interactions between age and occupation for this factor are all positive enough to compensate the negative effect of aging. However, this does not happen for less qualified types of occupation, nor for education.

The same exercise can be carried out for all possible combinations of sociodemographic characteristics included.

Results from Model 2 are shown in Table 4 (a detailed version of the table, Appendix A, including interaction effects and 95% confidence intervals, is available in online Appendix A).

According to Model 2 results, the crude effect of multimorbidity indicates that having declared three chronic conditions reduces the experience with the productive interactions and the new relational model, as well as the overall experience, compared to having a lesser number of conditions. The effect of multimorbidity is not significant on its own, but there are significant differences found when we interact the number of conditions with age, in particular between the older and younger groups. 

Regarding the productive interactions factor, having three conditions reduces experience the most for the younger and elder groups compared to the baseline. Results from the new relational model estimates show an aging effect, with differences in experience that increase with age no matter the number of conditions. Although this effect is higher the greater the number of conditions, these differences are only significant for respondents reporting three conditions. Calculations have been omitted in the main text. To compute differences, coefficients in Appendix A in the provided Appendix A were used. 

Experience with self-management decreases the most among the eldest respondents with two, three, and more than three conditions. Among respondents aged 65 and over, the largest difference found is for the 90 and over respondents and for the second youngest group with more than three conditions. No significant differences are found regarding the effect of multimorbidity on the experience with the self-management factor, though. 

Finally, experience overall decreases the most for the 65–74 age group among respondents with three chronic conditions compared to baseline. 

Results from Model 3 are shown in Table 5 below (a detailed version of the table, Appendix A, including interaction effects and 95% confidence intervals, is available in online Appendix A).

Patients with diabetes, asthma, malignancies, other mental disorders, anemia, or fibromyalgia reported better health care experience, denoted by their higher overall IEXPAC score and most of their partial factors. Individuals with hypertension and thrombosis only show better outcomes in self-management ability. Conversely, subjects with constipation, skin diseases, osteoarthritis, back or neck pain show lower scores. Additionally, notice that the model allows us to test the hypothesis of how different conditions, if acting as comorbidities (that is, the model can be restricted to the population suffering from one particular condition), would impact on patients’ experience, to see if there are differences between how one condition impacts on experience by itself, or when it is interacting with one or more conditions. To see it with an example, among patients with hypertension (for which the coefficient is 0.090) one could be interested in testing if the overall experience is significantly different depending on whether they have hypertension and cholesterol (0.090 + 0.069 = 0.159) or hypertension and diabetes (0.090 + 0.398 = 0.488). We contrasted the hypothesis of equality between both effects and observed that the difference is significant (−0.329, *p*-value < 0.001). This shows an example of how, while hypertension by itself does not have an impact on experience, interacting hypertension with other conditions can lead to a significant effect on experience. 

Finally, results from Model 4 are shown in Table 6 (a detailed version of the table, Appendix A, including interaction effects and 95% confidence intervals, is available in online Appendix A). 

Patients that reported a better quality of life (QoL), according to their EQ-5D-5L health status, also reported better health care experience. Such association was significant for all the analyzed factors except with the new relational model. A one-point increase in the QoL of a patient with any of the chronic conditions analyzed increases experience with the productive interactions and self-management factors, on average, by 0.563 and 0.742 points, respectively, and global experience by 0.535 points.

## 4. Discussion

This paper presents descriptive and regression analyses of the experience of care amongst people declaring any diagnosed chronic condition in the Population Health Survey 2018 in the Basque Country. 

Our results indicate that the analyzed IEXPAC factors are associated with gender, age, quality of life, chronic conditions, social class, and instruction level. Such findings are partially consistent with the published literature. 

Our population with chronic conditions reported on average good levels of patient experience with the health factors analyzed, with moderate–high mean scores in all IEXPAC factors, except for the factor that relates to the new relational model. In general, aging is also associated with worse experiences related to the new relational model. Such a finding is observed in our crude and adjusted results. All this is in line with previous evidence [20,40], our results also reinforcing the idea that elder individuals consider themselves misinformed about new technologies. This is consistent with other studies examining the relationship between internet access in health care and sociodemographic characteristics [41,42].

We also found that men, in general, report better health care experience values than women do. This fact is observed even adjusting by sociodemographic variables, quality of life, and multimorbidity. We did not find a significant difference by gender when including in the model the 39 chronic conditions, suggesting that for both genders chronic conditions also relate very differently with experience. We found differences by gender when we control by the number of comorbidities. This gender discrepancy may have a behavioral explanation. Other studies have demonstrated that women have more health problems, use more health services than men, and have lower self-reported health status than the opposite gender [43]. Note that a recently published study that focused on the same survey but studied the population with diabetes [20] found there were no differences between genders. Therefore, although chronic men and women may differ regarding health care experience, this effect needs to be tested for different chronic patient subgroups. 

Some significant associations between education and occupation variables with experience are found. Experience scores with all the experience factors analyzed are lower for people reporting lower educational levels, which is consistent with results from another recently published study [40]. Specifically, interacting age with education we observe significant differences between respondents with secondary-lower levels and those with primary education for all factors. Interestingly, for all factors except for the new relational model one, the negative effect over experience of having lower education levels is compensated when we interact age and education. It has been shown in the literature that lower education is associated with lesser participation in behaviors that imply the use of new technologies, such as communicating online with providers [44]. It is, therefore, not a surprise to find that those with lower education levels report worse experience with this factor. For occupation, similar to a previously published study of a population with diabetes [20], we do not observe a clear pattern when we interact it with age. Although this suggests there might not be differences between chronic patient subgroups in experience by occupation type, more research is encouraged to develop a better understanding regarding the effect of occupation and age on experience when seeking care. 

Multimorbidity is associated with patients´ experience, but contrary to other studies [45], remarkable effects arise only when we include age range interactions in our study. We also observe mixed results and the absence of a unique pattern. We only observe significant differences for patients reporting three comorbidities (compared to having one condition). A recently published study, which focused on the population with diabetes mellitus that responded to the IEXPAC questionnaire, concluded that there were no significant differences in IEXPAC scores due to the existence of comorbidities [40]. However, this study did not look for differences in experience scores when including interactions between age ranges and the number of comorbidities. Our study includes such interactions, and results are, thus, aligned with another recently published study [20] that found such differences in experience for multimorbidity when interacting this variable with age. More population-based studies, including of the full set of chronic conditions, are suggested to confirm that this result holds. 

Some chronic conditions are associated with differences in patients´ experience. In particular, our study shows patients with mental problems other than depression/anxiety, fibromyalgia, cancer, anemia, diabetes, and asthma showed better overall experience with care, while the ones with constipation, skin conditions and lower back pain reported lower global IEXPAC scores. Some findings reinforce previous findings in published literature. In our analyses, we observe that lower back pain significantly reduces chronic patients’ overall experience as well as experience with the new relational model factor, compared to the experience of respondents with other conditions. This finding is consistent with previous studies [26]. Literature has found that patients with lower back pain and similar types of conditions have noted areas of dissatisfaction with health care providers and perceived obstacles to care [46]. Other conditions, despite their low prevalence, also have a significant effect on the overall experience and with the self-management factor. Cancer patients reporting better experience may also be explained by low levels of stress and anxiety of these patients at radiotherapy, as a result of positive perceptions of these patients of higher climate of patient-centeredness at treatment and with the health professionals in charge of providing it, as discussed in a previous study [47]. The same paper also finds that safety issues and negative perceptions about other aspects, such as hospitality, may increase stress levels. Given that patients’ experiences can be used as an additional source for quality assessment [48] this suggests that, in our sample, the population with cancer, anemia and all other conditions that are associated with better experience, may have positive perceptions of the assistance provided to them. A recently published paper that analyzed the diabetic population that responded to the IEXPAC [20] concluded that there were no significant differences found in experience by chronic conditions. Note that if we were to restrict our chronic conditions model (Model 3) to the population of patients with diabetes, we would obtain identical results, as our diabetic population, control variables and interaction terms included are the same as the ones included in that study. In our study, we can test not only how each of the conditions, by themselves, impact experience, but also our Model 3, being linear, allows testing for the effect of any of the conditions when they interact with one condition chosen. This could be, thus, a model to use for practical implementation, to understand how experience is affected by any possible combination of chronic conditions.

Finally, regarding the effect of quality of life, we found that better quality of life is significantly associated with greater experience for two out of the three IEXPAC factors analyzed (the exception is the new relational model), as well as with the experience overall. Literature has identified HRQoL as the gap between expectations of health and experience of it [3]. According to our results, patients’ expectations are greater than experience by 0.5 points. Further research is encouraged to verify with other chronic patient populations if this result holds, as an indicator that there is room for improvement in the quality of the care delivered by a health system. Studying differences between groups of chronic patients would also be informative.

Information regarding the QoL for different conditions could be used in combination with the chronic conditions model for practical purposes such as decision-making. Because we also know the mean QoL associated with each condition in this population, one could combine both pieces of information to understand which conditions’ experience management would be more vulnerable according to their estimated impact on experience as well as on the QoL that is associated with those conditions. One could question causality, regarding the association between QoL and experience scores. Although our models cannot be used to establish causality, we know from previous literature that individuals seek care and use health services, which results in a QoL variation. To value experience with health services, one should first have made use of those. Therefore, it seems logical to think that it is the QoL associated with the health services provided what impacts on experience, and not the opposite. 

The results of this study have implications for health policy and health services’ management. Previous studies have demonstrated that better quality of care improves patients’ experience among those with chronic conditions [49]. Indicators of good quality of care include improved patient safety practices or clinical effectiveness in a wide range of diseases, designs, care settings, population groups, and lower utilization of unnecessary health care services [16]. A better knowledge of factors related to patient experience will give support to health care organizations to provide patients with better quality assistance. This fact could be even more relevant in vulnerable populations, such as patients with chronicity or multimorbidity. 

The principal strengths of this study are its large sample size (3981 individuals) and its representativeness. Our findings can be generalized to the wider population of patients with chronic conditions and their interaction with a health system. In addition, the IEXPAC instrument has several key advantages over existing instruments for assessing the patient experience of chronic care delivery [26]. First, this instrument can be used to detect changes over time and differences between patient subgroups. Second, in addition to information about patients’ clinical and risk factors, it is an important piece of information for decision-makers [33]. Third, it introduces a new focus on the interaction between patients and health care teams rather than just the experience with specific health care professionals, through the use of new technologies and patient–patient interactions [26]. In general, it incorporates a broader notion of integrated care, including social care and patient self-management [26]. Finally, model results are consistent, reflecting the robustness of our estimation method. In addition, our models are complementary. We not only provide separate estimates of the impact of the number of comorbidities and for the conditions but provide results so both pieces of information can be combined for interpretation.

The study does suffer from some limitations. First, this is a cross-sectional study and consequently does not allow us to establish causality in the observed relationships. Additionally, the information source for this study is the Basque Health Survey. We employed a list of 39 health problems (the ones included in the survey). This implies that no other conditions could be observed. The identification of chronic problems relies on the patients reporting diagnosis of a chronic condition. Although some degree of potential recall bias should be admitted, the prevalence rates of chronic conditions reported by our participants are almost identical to the estimates in the Basque Country from diagnosis registries. This results from a stratification protocol followed for their recruitment, which can be consulted elsewhere [50]. Second, we count conditions to control for multimorbidity, but there is no information regarding the severity of the progression of the diseases. The chronic conditions in the ESCAV survey are, clearly, a mix of simple conditions, risk factors, and more severe conditions. However, the classification of some conditions in one of these groups requires some degree of subjectivity. We, thus, decided to treat all chronic conditions in the survey equally, without making that distinction between more or less severe conditions or risk factors. Our third limitation refers to the use of IEXPAC factors as dependent variables for the regression analysis instead of IEXPAC independent items (several IEXPAC items form each of the factors, and each item is included in one factor only). We have conducted descriptive analysis for each IEXPAC item (see Figure 1), and our results were very similar to those of a previous study, which conducted a descriptive analysis using IEXPAC items, but then used IEXPAC factors as dependent variables for the regression analysis [36]. Although the context of that study was not the same—it was conducted with a non-representative sample of patients with inflammatory bowel disease and used the aggregated factors for the multivariate analyses—their analysis gives us confidence that this approach is valid and should give unbiased, relevant results. Fourth, other potential variables, such as risky behaviors, social support, or patient nationality were not requested, so they were not provided with the dataset and, therefore, are not included. Information related to the health care providers was not available either. The IEXPAC questionnaire included in the ESCAV survey asked for experience with health care services, but we cannot derive if they belonged to the public or private sectors. 

Further research could look at experiences disaggregating across several dimensions in an integrated care delivery system. Similar research using medical diagnosis and studying long-term, life-limiting chronic condition populations is encouraged. 

We believe that this paper contributes to a better understanding of the effect of socio-demographic, economic, and health-related characteristics on patients with chronic conditions with their care experiences. The lower levels of experience reported with the new relational model reflect the need for strategies that can help practitioners to harness the opportunities that new technologies and internet resources present for quality-of-care improvement [51,52,53]. Research using diagnostic information is encouraged to contrast our results with those from a clinically diagnosed population of patients with chronic conditions. 

## 5. Conclusions

This study identifies some differences in health care experience among those with chronic conditions in the Basque Country, as well as with health-related factors. There seems to be a negative association between health care experience and multimorbidity, as well as between experience and some chronic conditions. Experience with health care factors also improves with better quality of life. Measuring patient experience of the interaction with a care system provides useful information for health care management, although this cannot substitute for analysis of this topic at the health care provider or in specific settings (hospitals, nursing homes, etc.). The availability of other studies with a similar focus can allow the comparison of the performance of different health care models.

## Figures and Tables

**Figure 1 healthcare-09-01005-f001:**
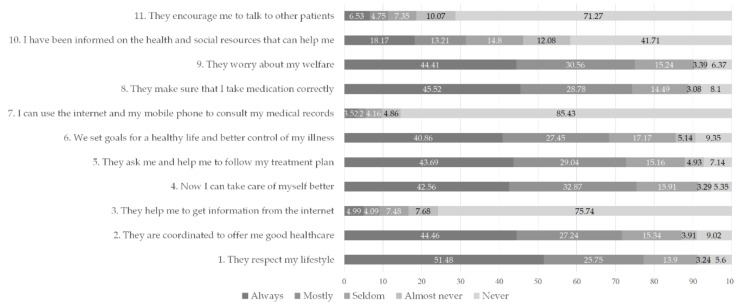
Distribution of patients’ responses to IEXPAC items. Numbers in bars represent the percentage of respondents who responded to each option. Missing answers (*n* = 1, 0.03%) were removed to make the figure.

**Table 1 healthcare-09-01005-t001:** Sociodemographic variables (*). Descriptive statistics.

Sociodemographic Variables	Obs. (a)	Variable Mean (s.e)	EQ-5Dindex Mean (s.e)	Factor 1: INTER Mean (s.e)	Sig. (b)	Factor 2: NEW Mean (s.e)	Sig. (c)	Factor 3: SELF Mean (s.e)	Sig. (d)	OVERALL IEXPAC Mean (s.e)	Sig. (e)
Overall sample	3891		0.867 (0.003)	7.556 (0.040)		1.277 (0.031)		6.542 (0.039)		5.468 (0.031)	
Gender	3891										
Women	2169	0.557 (0.497)	0.843 (0.004)	7.487 (2.596)	**	1.245 (1.939)	*****	6.469 (2.481)		5.414 (1.975)	**
Men	1722	0.443 (0.497)	0.898 (0.004)	7.644 (2.399)		1.319 (2.028)		6.596 (2.409)		5.538 (1.925)	
Age	3891	62.91 (16.32)									
15–24	86	0.022 (0.147)	0.947 (0.010)	7.565 (1.962)		2.064 (2.293)	*****	6.395 (2.113)	*	5.640 (1.742)	**
25–44	462	0.119 (0.324)	0.913 (0.007)	7.100 (2.554)		1.659 (2.234)	*****	5.894 (2.402)	***	5.178 (2.011)	
45–64	1393	0.358 (0.479)	0.890 (0.004)	7.248 (2.568)	***	1.484 (2.154)	*******	6.283 (2.459)	***	5.325 (2.023)	***
65–74	948	0.244 (0.429)	0.895 (0.004)	7.667 (2.529)	***	1.108 (1.811)	*******	6.729 (2.421)	**	5.537 (1.948)	**
75–89	913	0.235 (0.424)	0.797 (0.008)	8.095 (2.297)		0.916 (1.622)		6.966 (2.386)		5.726 (1.794)	
90+	89	0.023 (0.15)	0.625 (0.033)	8.048 (2.678)		0.815 (1.443)		7.015 (2.709)		5.700 (2.012)	
Net monthly income	2142										
No income	1	0 (0.022)	0.340 (NA)	6.250 (.)		0 (.)		6.250 (.)		4.545 (.)	
<=500 €	30	0.014 (0.118)	0.865 (0.026)	7.396 (2.373)		1.028 (1.270)		6.417 (2.344)		5.303 (1.766)	
501–1000 €	431	0.201 (0.401)	0.802 (0.011)	7.946 (2.212)		0.979 (1.760)		6.895 (2.266)		5.664 (1.721)	
1001–1500 €	600	0.28 (0.449)	0.856 (0.008)	7.773 (2.421)		1.165 (1.848)		6.693 (2.390)		5.578 (1.874)	
1501–2000 €	408	0.19 (0.393)	0.901 (0.007)	7.431 (2.680)	**	1.185 (1.962)		6.498 (2.569)	*	5.388 (2.067)	
2001–2500 €	294	0.137 (0.344)	0.911 (0.008)	7.188 (2.762)		1.247 (1.895)	*******	6.135 (2.577)		5.185 (2.067)	
2501–3500 €	263	0.123 (0.328)	0.904 (0.009)	7.391 (2.667)		1.616 (2.291)	*******	6.355 (2.500)		5.439 (2.088)	*
3501–5000 €	96	0.045 (0.207)	0.919 (0.012)	7.695 (2.461)		2.483 (2.553)		6.543 (2.418)		5.855 (2.033)	
5001–7000 €	14	0.007 (0.081)	0.930 (0.035)	7.545 (2.832)		2.321 (2.945)	*****	5.982 (3.259)		5.552 (2.510)	
7001–10,000 €	3	0.001 (0.037)	1 (0)	9.167 (0.955)		0 (0)	**NA**	5.833 (0.955)		5.455 (0.455)	
More than 10,000 €	2	0.001 (0.031)	1 (0)	10 (0)		0 (0)		8.125 (0.884)		6.591 (0.321)	
Level of education	3891										
Primary education	1287	0.331 (0.471)	0.793 (0.020)	7.824 (2.669)	***	1.290 (1.924)	*******	6.849 (2.592)	***	5.688 (2.077)	***
Secondary—lower	790	0.203 (0.402)	0.785 (0.017)	7.531 (2.205)		0.748 (1.600)	*******	6.468 (2.169)		5.294 (1.674)	
Secondary—upper	1243	0.319 (0.466)	0.826 (0.008)	7.415 (2.455)		1.433 (2.084)	*******	6.427 (2.374)	***	5.425 (1.920)	
Tertiary	571	0.147 (0.354)	0.895 (0.005)	7.296 (2.617)		1.643 (2.191)		6.086 (2.561)		5.314 (2.056)	
Occupation	3884										
Managers I	577	0.149 (0.356)	0.890 (0.007)	7.608 (2.561)		1.388 (2.058)	*******	6.425 (2.480)		5.482 (1.961)	
Managers II	232	0.06 (0.237)	0.886 (0.011)	7.398 (2.507)		1.695 (2.424)	*******	6.444 (2.558)		5.496 (2.088)	
Intermediate	176	0.045 (0.208)	0.858 (0.015)	7.148 (2.680)	**	1.236 (1.828)		6.289 (2.549)	*	5.223 (2.027)	**
Semi-qualified	923	0.238 (0.426)	0.870 (0.006)	7.637 (2.547)		1.326 (2.036)	*****	6.640 (2.580)		5.553 (2.016)	
Non-qualified	1976	0.509 (0.5)	0.859 (0.004)	7.555 (2.465)		1.179 (1.877)		6.528 (2.356)		5.443 (1.900)	

(a) The difference between the sample size (*n* = 3891) and the number of responses for a category/variable (Obs.) represents the number of missing responses for that category/variable; (b–e) Significance (Sig.) of group differences of means of the IEXPAC factors for each sociodemographic variable. Tests are carried out for the closest pairs of categories within each variable only (e.g., 15–24 vs. 25–44, 25–44 vs. 45–64…), but not between alternate categories (e.g., 15–24 vs. 45–64); s.e.: standard error. All variable subgroups are dummy variables, taking values 0 or 1. Therefore, the variable mean can easily be converted (% = variable mean × 100) into the percentage of individuals in each subgroup/category (* *p* < 0.1, ** *p* < 0.05, *** *p* < 0.001).

**Table 2 healthcare-09-01005-t002:** Health-related variables. Descriptive statistics.

Health-Related Variables	Obs.	Age Mean	Variable Mean (s.e)	EQ-5Dindex Mean (s.e)	Factor 1: INTER Mean (s.e)	Sig. (a)	Factor 2: NEW Mean (s.e)	Sig.	Factor 3: SELF Mean (s.e)	Sig.	OVERALL IEXPAC Mean (s.e)	Sig.
Chronic conditions	3891	62.91	1 (NA)	0.868 (0.198)	7.556 (2.511)		1.277 (1.979)		6.524 (2.449)		5.468 (1.953)	
Hypertension	1606	70.99	0.413 (0.492)	0.849 (0.005)	7.806 (2.453)	***	1.119 (1.843)	***	6.796 (2.369)	***	5.615 (1.892)	***
Cholesterol (high)	1179	62.29	0.303 (0.46)	0.846 (0.006)	7.821 (2.319)		1.082 (1.824)		6.783 (2.293)		5.606 (1.793)	
Osteoarthritis	706	68.55	0.181 (0.385)	0.728 (0.009)	7.474 (2.448)		0.845 (1.585)		6.472 (2.377)		5.302 (1.802)	
Lower back pain	590	64.11	0.152 (0.359)	0.730 (0.010)	7.398 (2.176)		0.775 (1.499)		6.232 (2.184)		5.168 (1.613)	
Diabetes	555	70.86	0.143 (0.35)	0.815 (0.010)	8.176 (2.048)		1.222 (1.937)		7.193 (2.085)		5.922 (1.658)	
Other cardiovascular	419	74.22	0.108 (0.31)	0.789 (0.011)	7.879 (2.403)		1.064 (1.806)		6.868 (2.485)		5.653 (1.902)	
Neck pain	385	64.4	0.099 (0.299)	0.697 (0.013)	7.433 (2.104)		0.701 (1.396)		6.179 (2.059)		5.141 (1.515)	
Thyroids	371	62.61	0.095 (0.294)	0.864 (0.010)	7.465 (2.750)		1.274 (1.998)		6.504 (2.496)		5.427 (2.056)	
Insomnia	346	70.13	0.089 (0.285)	0.715 (0.015)	7.493 (2.630)		0.855 (1.632)		6.438 (2.436)		5.299 (1.926)	
Allergy	275	46.89	0.071 (0.256)	0.882 (0.012)	7.168 (2.388)		1.261 (1.821)		6.043 (2.338)		5.148 (1.832)	
Asthma	270	56.75	0.069 (0.254)	0.816 (0.015)	7.644 (2.471)		1.451 (2.237)		6.544 (2.410)		5.555 (2.006)	
Deafness	259	73.07	0.067 (0.249)	0.737 (0.016)	7.867 (1.992)		0.714 (1.563)		6.742 (2.061)		5.507 (1.505)	
Osteoporosis	242	72.80	0.062 (0.242)	0.711 (0.019)	7.477 (2.383)		0.937 (1.614)		6.493 (2.312)		5.335 (1.822)	
Cardiovascular	241	71.47	0.062 (0.241)	0.711 (0.017)	7.876 (2.073)		0.764 (1.610)		6.546 (2.171)		5.453 (1.591)	
Varicose veins (legs)	232	68.77	0.06 (0.237)	0.752 (0.016)	7.713 (1.980)		0.528 (1.194)		6.503 (2.009)		5.313 (1.440)	
Other	212	60.18	0.054 (0.227)	0.851 (0.014)	7.453 (2.535)		1.384 (2.140)		6.557 (2.389)		5.472 (1.927)	
Skin conditions	184	55.90	0.047 (0.212)	0.806 (0.018)	7.137 (2.483)		0.992 (1.487)		5.971 (2.347)		5.037 (1.786)	
Depression	177	65.49	0.045 (0.208)	0.605 (0.022)	7.429 (2.531)		1.243 (1.995)		6.631 (2.511)		5.452 (2.000)	
Anxiety	175	61.34	0.045 (0.207)	0.661 (0.020)	7.150 (2.628)		1.333 (1.945)		6.361 (2.581)		5.277 (2.031)	
Migraine	169	55.56	0.043 (0.204)	0.724 (0.020)	7.223 (2.296)		1.011 (1.672)		6.072 (2.256)		5.110 (1.727)	
Other mouth	161	62.46	0.041 (0.199)	0.706 (0.022)	7.007 (2.312)		0.932 (1.746)		6.211 (2.228)		5.061 (1.711)	
Caries	159	57.36	0.041 (0.198)	0.747 (0.213)	7.248 (2.300)		0.839 (1.510)		6.002 (2.256)		5.047 (1.703)	
Peptic Ulcer condition	159	58.13	0.041 (0.198)	0.824 (0.017)	6.981 (2.792)		1.515 (2.148)		6.053 (2.670)		5.153 (2.142)	
Prostate	152	73.95	0.039 (0.194)	0.859 (0.015)	8.141 (1.785)		0.872 (1.668)		6.928 (2.148)		5.718 (1.522)	
Hemorrhoids	126	64.80	0.032 (0.177)	0.706 (0.027)	7.341 (2.133)		0.774 (1.647)		6.399 (2.164)		5.207 (1.639)	
Cancer	126	66.63	0.032 (0.177)	0.812 (0.021)	7.862 (2.574)		1.951 (2.565)		7.029 (2.650)		5.947 (2.205)	
Cataracts	117	75.24	0.03 (0.171)	0.744 (0.026)	7.917 (1.873)		0.684 (1.475)		6.725 (1.981)		5.511 (1.404)	
COPD	112	67.98	0.029 (0.167)	0.758 (0.026)	7.690 (2.834)		1.213 (2.092)		6.791 (2.639)		5.597 (2.190)	
Incontinence	107	74.39	0.027 (0.164)	0.532 (0.034)	7.593 (2.485)		0.857 (1.981)		6.519 (2.398)		5.365 (1.853)	
Kidney conditions	105	70	0.027 (0.162)	0.708 (0.027)	7.482 (2.726)		1.119 (1.884)		6.649 (2.509)		5.444 (1.976)	
Blindness	102	68.45	0.026 (0.16)	0.783 (0.026)	7.426 (2.530)		1.078 (1.907)		6.532 (2.456)		5.370 (1.903)	
Dementia	100	76.85	0.026 (0.158)	0.573 (0.034)	7.838 (2.767)		1.367 (1.987)		6.675 (3.036)		5.650 (2.281)	
Constipation	93	70.07	0.024 (0.153)	0.556 (0.035)	7.245 (2.220)		0.609 (1.217)		5.907 (2.168)		4.949 (1.583)	
Anemia	78	63.89	0.02 (0.14)	0.737 (0.034)	8.069 (1.830)		1.218 (1.850)		6.931 (1.974)		5.787 (1.444)	
Thrombosis	74	73.52	0.019 (0.137)	0.705 (0.034)	8.015 (2.258)		0.901 (1.471)		7.272 (2.134)		5.805 (1.634)	
Other mental	72	56.05	0.019 (0.135)	0.762 (0.033)	7.969 (2.001)		2.106 (2.624)		7.188 (2.392)		6.086 (1.953)	
Fibromyalgia	60	61.26	0.015 (0.123)	0.674 (0.031)	7.417 (2.561)		1.986 (2.302)		6.854 (2.134)		5.731 (1.919)	
AMI	52	75.57	0.013 (0.115)	0.764 (0.033)	7.993 (2.091)		1.234 (1.914)		6.935 (2.094)		5.765 (1.635)	
Diabetic foot	11	75.54	0.003 (0.053)	0.494 (0.093)	8.068 (1.711)		1.742 (2.156)		7.557 (2.491)		6.157 (1.845)	
Number of chronic conditions	3886											
1	1505	55.93	0.387 (0.487)	0.937	8.351 (1.965)	***	1.250 (1.901)	**	7.198 (2.150)		5.995 (1.604)	
2	924	64.07	0.238 (0.426	0.903	8.163 (8.071)		1.740 (2.171)		7.495 (2.031)		6.168 (1.781)	
3	578	67.99	0.149 (0.356)	0.867	8.071 (2.115)	***	1.361 (2.153)		7.090 (2.143)		5.884 (1.744)	
>3	879	70.43	0.226 (0.418)	0.708	8.194 (2.002)		0.915 (1.654)		7.108 (2.063)		5.814 (1.564)	
EQ-5Dindex	3891		0.868 (0.198)									

COPD: chronic obstructive pulmonary disease; AMI: acute myocardial infarction; NA: Not applicable; s.e: standard error. (a) Significance has been tested between the difference in experience (measured by score means difference) for each factor, between the most and least prevalent conditions (hypertension and diabetic foot) (* *p* < 0.1, ** *p* < 0.05, *** *p* < 0.001).

**Table 3 healthcare-09-01005-t003:** Model 1—WLS results. Differences in health care experience among patients with self-declared chronic conditions. The effect of sociodemographic and economic characteristics.

Variable	Category	Factor 1: INTER Coef.	Factor 2: NEW Coef.	Factor 3: SELF Coef.	OVERALL IEXPAC Coef.
Gender Baseline: Women	Men	0.214 **	0.022	0.168 **	0.145 **
Age groups	25–44	−1.373 *	−1.022	−1.080	−1.171
Baseline: 15–24	45–64	−0.808	−1.249	−0.264	−0.730
	65–74	−0.715	−1.659	−0.338	−0.835
	75–89	−0.097	−1.848 *	0.403	−0.393
	>=90	−1.121	−2.464 **	−1.186	−1.511
Occupation	Managers II	−0.355	−2.631 ***	−0.693	−1.098 **
Baseline: Managers I	Intermediate	−0.660	−0.169	1.561 **	0.282
	Semi-qualified	0.616	1.452 *	0.974	0.974 *
	Non-qualified	0.094	−0.826	0.393	−0.048
Education	Secondary-lower	−1.951 **	−2.070 **	−2.425 **	−2.156 **
Baseline: Primary	Secondary-upper	−1.041 **	−0.404	−0.786	−0.775
	Tertiary	−0.361	−0.636	−0.656	−0.544
Interactions	Occupation#Age	YES	YES	YES	YES
	Education#Age	YES	YES	YES	YES
Constant term	Constant	8.369 ***	3.013 **	6.936 ***	6.387 ***
Goodness-of-fit	R-squared	0.034	0.055	0.039	0.027
	BIC	18,445.536	16,501.215	18,223.872	16,523.464
Sample Size (¥)	*n*	3883	3883	3883	3883

* *p* < 0.1, ** *p* < 0.05, *** *p* < 0.001; Coef.: Regression coefficient; BIC: Bayesian information criterion; the presented model is corrected from heteroscedasticity using Eicker–Huber–White standard errors. A detailed table including results for the interactions and 95% confidence intervals (95% CI) is available in Appendix A (Appendix A). ¥: Missing responses excluded for the analyses. # Represents the interaction between two variables.

**Table 4 healthcare-09-01005-t004:** Model 2—WLS results. Differences in health care experience among patients with self-declared chronic conditions. The effect of chronic multimorbidity.

Independent Variables	Category	Factor 1: INTER Coef.	Factor 2: NEW Coef.	Factor 3: SELF Coef.	OVERALL IEXPAC Coef.
Gender Baseline: Women	Men	0.210 **	0.013	0.160 **	0.138 **
Age	25–44	−1.493 *	−1.087	−1.111	−1.243
Baseline: 15–24	45–64	−0.890	−1.201	−0.305	−0.762
	65–74	−1.059 *	−1.627	−0.608	−1.050
	75–89	−0.0195	−1.648	0.548	−0.257
	>=90	−2.066	−3.026 **	−2.399	−2.449 **
Education	Secondary—lower	−2.165 **	−2.225 **	−2.491 **	−2.300 ***
Baseline: Primary	Secondary—upper	−1.169 **	−0.446	−0.826	−0.847
	Tertiary	−0.252	−0.925	−0.703	−0.600
Occupation	Managers II	−0.360	−2.656 ***	−0.684	−1.104 **
Baseline: Managers I	Intermediate	−0.577	−0.188	1.583 **	0.315
	Semi-qualified	0.676	1.398 *	0.961	0.977 *
	Non-qualified	0.303	−0.686	0.451	0.087
Number of diseases	2	−0.636	0.286	−0.0676	−0.178
Baseline: 1	3	−1.528 *	−1.870 ***	−0.520	−1.255 **
	+3	0.341	0.307	0.322	0.325
Interactions	Age#Number of conditions	YES	YES	YES	YES
	Age#Education	YES	YES	YES	YES
	Age#Occupation	YES	YES	YES	YES
Constant term	Constant	8.506 ***	3.089 **	6.975 ***	6.472 ***
Goodness-of-fit	R-squared	0.0392	0.0630	0.0439	0.0326
	BIC	18,549.184	16,597.597	18,324.712	16,624.089
Sample size (¥)	*n*	3878	3878	3878	3878

* *p* < 0.1, ** *p* < 0.05, *** *p* < 0.001; Coef.: Regression coefficient; BIC: Bayesian information criterion; the presented model is corrected from heteroscedasticity using Eicker–Huber–White standard errors. A detailed table including results for the interactions and 95% confidence intervals (95% CI) is available in Appendix A (Appendix A). ¥: Missing responses excluded for the analyses. # Represents the interaction between two variables.

**Table 5 healthcare-09-01005-t005:** Model 3—WLS results. Differences in health care experience among patients with self-declared chronic conditions. The effect of chronic conditions.

Independent Variables	Category	Factor 1: INTER	Factor 2: NEW	Factor 3: SELF	OVERALL IEXPAC
		Coef.	Coef.	Coef.	Coef.
Gender Baseline: Women	Men	0.075	0.013	0.059	0.052
Age ranges.	25–44	−1.220	−0.888	−0.942	−1.028
Baseline: 15–24	45–64	−0.705	−1.063	−0.163	−0.606
	65–74	−0.787	−1.469	−0.423	−0.841
	75–89	−0.164	−1.644	0.357	−0.378
	>=90	−1.160	−2.141 *	−1.251	−1.460
Occupation	Managers II	−0.136	−2.470 ***	−0.537	−0.918 **
Baseline: Managers I	Intermediate	−0.520	−0.171	1.725 **	0.392
	Semi-qualified	0.739	1.324 *	1.011	0.998 *
	Non-qualified	0.174	−0.779	0.424	0.005
Education	Secondary-lower	−1.774 **	−1.849 *	−2.169 **	−1.938 **
Baseline: Primary	Secondary-upper	−0.871	−0.180	−0.525	−0.557
	Tertiary	−0.126	−0.270	−0.240	−0.207
Chronic conditions	Hypertension	0.109	−0.027	0.160 *	0.090
	Cholesterol (high)	0.144	−0.102	0.123	0.069
	Osteoarthritis	−0.119	−0.158 *	−0.007	−0.089
	Lower back pain	−0.094	−0.269 **	−0.184	−0.174 *
	Diabetes	0.485 ***	0.120	0.521 ***	0.398 ***
	Other cardiovascular	0.021	0.063	0.107	0.064
	Neck pain	0.015	−0.169 *	−0.240 *	−0.128
	Thyroids	−0.009	−0.009	0.060	0.016
	Insomnia	−0.079	−0.078	−0.095	−0.085
	Allergy	−0.146	−0.245 **	−0.113	−0.161
	Asthma	0.216	0.230	0.232	0.226 *
	Deafness	0.155	−0.151	0.132	0.063
	Osteoporosis	−0.175	−0.039	−0.144	−0.127
	Cardiovascular	0.203	−0.129	−0.112	−0.002
	Varicose veins (legs)	0.198	−0.259 **	0.113	0.042
	Other	−0.030	0.024	0.093	0.030
	Skin	−0.306	−0.287 **	−0.351 **	−0.317 **
	Depression	0.036	0.057	0.188	0.097
	Anxiety	−0.289	0.228	−0.017	−0.049
	Migraine	−0.043	−0.110	−0.101	−0.082
	Other mouth	−0.528 **	0.036	−0.158	−0.239
	Caries	0.075	−0.163	−0.165	−0.077
	Peptic Ulcer disease	−0.441 *	0.179	−0.297	−0.220
	Prostate	0.311 *	−0.280 *	0.146	0.090
	Hemorrhoids	−0.141	0.033	0.099	−0.006
	Cancer	0.254	0.742 ***	0.504 **	0.478 **
	Cataracts	0.053	−0.031	0.009	0.014
	COPD	0.060	0.034	0.206	0.106
	Incontinence	−0.200	−0.024	−0.247	−0.169
	Kidney conditions	−0.310	0.162	−0.056	−0.089
	Blindness	−0.208	−0.043	−0.070	−0.113
	Dementia	0.017	0.322	−0.066	0.070
	Constipation	−0.325	−0.144	−0.609 **	−0.379 **
	Anemia	0.556 **	0.212	0.523 **	0.450 **
	Thrombosis	0.292	−0.036	0.649 **	0.333
	Other mental	0.550 **	0.548 *	0.773 **	0.631 **
	Fibromyalgia	0.154	0.878 **	0.612 **	0.518 **
	AMI	0.118	0.221	0.094	0.137
	Diabetic foot	−0.180	0.780	0.391	0.289
Interactions	Age#Occupation	YES	YES	YES	YES
	Age#Education	YES	YES	YES	YES
Constant term	Constant	8.266 ***	2.882 **	6.741 ***	6.243 ***
Goodness-of-fit	R-squared	0.051	0.082	0.061	0.049
	BIC	18,697.108	16,711.820	18,455.281	16,755.162
Sample size (¥)	*n*	3883	3883	3883	3883

* *p* < 0.1, ** *p* < 0.05, *** *p* < 0.001; Coef.: Regression coefficient; COPD: chronic obstructive pulmonary disease; AMI: acute myocardial infarction. BIC: Bayesian information criterion; the presented model is corrected from heteroscedasticity using Eicker–Huber–White standard errors. A detailed table including results for the interactions and 95% confidence intervals (95% CI) is available in Appendix A (Appendix A). ¥: Missing responses excluded for the analyses. # Represents the interaction between two variables.

**Table 6 healthcare-09-01005-t006:** Model 4—WLS results. Differences in health care experience among patients with self-declared chronic conditions. The effect of quality of life is represented by the EQ-5Dindex.

Independent Variables	Category	Factor 1: INTER	Factor 2: NEW	Factor 3: SELF	OVERALL IEXPAC
		Coef.	Coef.	Coef.	Coef.
Gender Baseline: Women	Men	0.141 *	0.012	0.178 **	0.119 *
Age	25–44	−1.022	−1.000	−1.297	−1.116
Baseline: 15–24	45–64	−0.225	−1.233	−0.756	−0.693
	65–74	−0.311	−1.649	−0.680	−0.810
	75–89	0.494	−1.812 *	0.022	−0.306
	>=90	−0.988	−2.386 **	−0.861	−1.323
Occupation	Managers II	−0.700	−2.634 ***	−0.365	−1.106 **
Baseline: Managers I	Intermediate	1.570 **	−0.166	−0.649	0.290
	Semi-qualified	0.999	1.461 *	0.647	0.997
	Non-qualified	0.399	−0.824	0.102	−0.043
Education	Secondary-lower	−2.427 **	−2.071 **	−1.954 **	−2.158 **
Baseline: Primary	Secondary-upper	−0.791	−0.406	−1.047 **	−0.779
	Tertiary	−0.670	−0.641	−0.379	−0.556
Interactions	Occupation#Age	YES	YES	YES	YES
	Education#Age	YES	YES	YES	YES
QoL measure	EQ-5D index	0.563 **	0.222	0.742 **	0.535 **
Constant term	Constant	6.412 ***	2.807 **	7.679 ***	5.890 ***
Goodness-of-fit	R-squared	0.040	0.056	0.037	0.029
	BIC	18,224.702	16,507.682	18,441.584	16,521.305
Sample size (¥)	*n*	3883	3883	3883	3883

** p* < 0.1, *** p* < 0.05, **** p* < 0.001; Coef.: Regression coefficient; QoL: Quality of Life; BIC: Bayesian information criterion; the presented model is corrected from heteroscedasticity using Eicker–Huber–White standard errors. A detailed table including results for the interactions and 95% confidence intervals (95% CI) is available in Appendix A (Appendix A). ¥: Missing responses excluded for the analyses. # Represents the interaction between two variables.

## Data Availability

The datasets generated and/or analysed during the current study are available from the corresponding author on reasonable request.

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
