# Peer review of "Inequalities in Health Care Experience of Patients with Chronic Conditions: Results from a Population-Based Study"

_healthcare, 2021, doi:10.3390/healthcare9081005_

Round 1
Reviewer 1 Report
The paper is in accordance with the journal, is written in a rigorous and academically correct style. The authors present an analysis of the elements that may be influencing the experience of Basque patients with the health system according to their chronic pathologies.
The introduction justifies the Basque healthcare system as well as the difference between the traditional measure of patient satisfaction versus the idea of user experience. Perhaps in the introduction they could justify why they have continued to use the Triple Aim when they are now talking about the Quadruple Aim.
The method is well developed, only two pieces of information that were not entirely clear to the reviewer could be better specified:
2.2. Design and Working Sample
It is not clear where the study sample has finally come from. Whether from the 12995 families or from the 8036 individuals, mentioned in the previous section.
2.3.2 Independent variables
They mention the classification they have used for age but do not specify it as they do for occupation, for example.
Finally, in the discussion, they expand on the benefits of their findings for the Basque, Spanish or world health system, bearing in mind that resources are finite and that efforts must be focused on key points in the system.
Reviewer 2 Report
In this manuscript the authors aim to find possible inequalities in which patients with chronic conditions experience the healthcare delivery process using data from the 2018 Basque population-based Health Survey. Data were analyzed for 38 different chronic conditions separately, or for the number of chronic conditions per patient, and for QoL, on the aspects productive interactions (INTER), new ways of interaction (NEW) and self-management ability (SELF), either separately or combined (OVERALL). No remarkable inequalities for these aspects were found. Having diabetes was associated with better appreciation of healthcare experience whereas having lower back pain was associated with poorer appreciation. Previously, the authors published a very similar analysis on a subset from the current study population, the 555 respondents having diabetes. In that paper, having additional comorbidities significantly decreased experience with all healthcare services.
Major:
- In the discussion, please relate the results of the current study with those of the previous study on the subset of respondents having diabetes (ref. #13).
- It is hard to grasp the most important results from this study. Usually, authors start the discussion with their most important results. Instead, the authors start with a lengthy discussion on the strengths and limitations of their study. The real discussion on the results only starts from line 426.
- A massive amount of data is generated, but what the possible implications are for healthcare delivery processes is unclear. The statements on lines 455-457 and 466-467 remain rather vague and could likely have been made well without the current study. Please elaborate more specifically on the possible implications of the present results.
- A higher QoL is associated with better experience with healthcare services. Please speculate on possible causality and order of causality.
Minor:
- abstract line 20: please include number of respondents: 3981 out of … respondents;
- Line 231: Isn’t the new relational model Factor 2 iso factor 3?
- Line 253: lower than what?
- Line 285: please explain where +2.659 comes from. Similarly line 296 and 308.
- Line 287: please explain what is meant with …, for which would be lower in 0.878 points.
- Lines 390/393: please elaborate on the statement that from the Model 2 results (Table 4) one can derive that the number of comorbidities is more important that the specific disease.
- Please check BIC values in table 4.
Reviewer 3 Report
This cross sectional study is really interesting and give a new perspective of health related perceptions in Basque population. It is well designed, correctly conducted, there are no ethical problems and results are interesting. However I have some considerations to put on attention to the authors to modify the manuscript before been accepted for pubblication. 1) All the questions of the IEXPAC questioner should be mentioned in the manuscript. 2) Figure 1 is quite unreadable. 3) The authors stress rightly that it is a population-based study: as a matter of fact all the chronic conditions listed are not distinguished in simple chronic conditions (i.e. back pain, incontinence, constipation) , risk factors (hypertension and hypercholesterolemia) and disease (COPD, diabetes,). So far what it is experienced by the patient is a mixture of everthing inclusive diseses if present. The authors shoul better specify what is comorbidity or multimorbidity that is a medical term not, again, what it is experienced by a patient. 4) would it possible to have a median number of medicaments prescribed to patients or at least for adults (from 65 or older)?
Round 2
Reviewer 2 Report
The authors have addressed my comments more than adequately. One remark remains: please check the BIC numbers in Table 4, as either the comma's or decimal points seem misplaced.
Author Response
Thank you for your quick feedback and for your comment. We are happy to see that we have adequately responded to your concerns in previous revision.
We have now realised what was the error and corrected the BIC numbers in Table 4.